# The Prevalence of Metallo-Beta-Lactamese-(MβL)-Producing *Pseudomonas aeruginosa* Isolates in Brazil: A Systematic Review and Meta-Analysis

**DOI:** 10.3390/microorganisms11092366

**Published:** 2023-09-21

**Authors:** Pabllo Antonny Silva Dos Santos, Marcos Jessé Abrahão Silva, Maria Isabel Montoril Gouveia, Luana Nepomuceno Gondim Costa Lima, Ana Judith Pires Garcia Quaresma, Patrícia Danielle Lima De Lima, Danielle Murici Brasiliense, Karla Valéria Batista Lima, Yan Corrêa Rodrigues

**Affiliations:** 1Program in Parasitic Biology in the Amazon Region (PPGBPA), State University of Pará (UEPA), Belém 66087-662, PA, Brazil; antonnypabllo@gmail.com (P.A.S.D.S.); luanalima@iec.gov.br (L.N.G.C.L.); patriciadanielle@bol.com.br (P.D.L.D.L.); daniellemurici@iec.gov.br (D.M.B.); karlalima@iec.gov.br (K.V.B.L.); 2Bacteriology and Mycology Section, Evandro Chagas Institute (SABMI/IEC), Ministry of Health, Ananindeua 67030-000, PA, Brazil; isabelmontoril13@gmail.com (M.I.M.G.); anaquaresma@iec.gov.br (A.J.P.G.Q.); 3Program in Epidemiology and Health Surveillance (PPGEVS), Evandro Chagas Institute (IEC), Ministry of Health, Ananindeua 67030-000, PA, Brazil; 4Department of Natural Science, State University of Pará (DCNA/UEPA), Belém 66050-540, PA, Brazil

**Keywords:** *Pseudomonas aeruginosa*, meta-analysis, Brazil, MΒL, *bla*
_SPM-1_

## Abstract

The purpose of the current study is to describe the prevalence of *Pseudomonas aeruginosa* (PA)-producing MβL among Brazilian isolates and the frequency of *bla*_SPM-1_ in MβL-PA-producing isolates. From January 2009 to August 2023, we carried out an investigation on this subject in the internet databases SciELO, PubMed, Science Direct, and LILACS. A total of 20 papers that met the eligibility requirements were chosen by comprehensive meta-analysis software v2.2 for data retrieval and analysis by one meta-analysis using a fixed-effects model for the two investigations. The prevalence of MβL-producing *P. aeruginosa* was 35.8% or 0.358 (95% CI = 0.324–0.393). The studies’ differences were significantly different from one another (x^2^ = 243.15; *p* < 0.001; I^2^ = 92.18%), so they were divided into subgroups based on Brazilian regions. There was indication of asymmetry in the meta-analyses’ publishing bias funnel plot; so, a meta-regression was conducted by the study’s publication year. According to the findings of Begg’s test, no discernible publishing bias was found. *bla*_SPM-1_ prevalence was estimated at 66.9% or 0.669 in MβL-PA isolates (95% CI = 0.593–0.738). The analysis of this one showed an average heterogeneity (x^2^ = 90.93; *p* < 0.001; I^2^ = 80.20%). According to the results of Begg’s test and a funnel plot, no discernible publishing bias was found. The research showed that MβL-*P. aeruginosa* and SPM-1 isolates were relatively common among individuals in Brazil. *P. aeruginosa* and other opportunistic bacteria are spreading quickly and causing severe infections, so efforts are needed to pinpoint risk factors, reservoirs, transmission pathways, and the origin of infection.

## 1. Introduction

The emergence and active spread of carbapenem-resistant *Pseudomonas aeruginosa* (CR-PA) strains are a major concern, especially in healthcare settings, as this class of broad-spectrum antibiotics is often used as a last resort to treat infections caused by multidrug-resistant (MDR) or extensive (XDR) isolates [1,2,3]. Cases of poor prognosis, complicated infections, and high mortality rates have been demonstrated to CR-PA infections, such as bloodstream infections (BSIs), ventilator-associated pneumonia (VAP), and severe burns in hospitalized patients, especially among those in intensive care units (ICUs) [4,5,6,7]. Data from the Antimicrobial Testing Leadership and Surveillance program demonstrate that the highest CR rates are in the Middle East, followed by South America, in which Brazil appears with 16.4% among 116 evaluated isolates [8]. Such increasing resistance trends have also been verified in other neighboring countries, including Bolivia and Peru [2].

Several resistance mechanisms lead to non-susceptibility phenotypes in *P. aeruginosa* strains, including the production of carbapenemases, changes in outer membrane permeability, and overexpression of efflux pumps that can expel the antibiotic from the cell. Such phenomena generally occur in selective pressure environments along with the ability of *P. aeruginosa* to form biofilms that can provide a protective environment for antibiotic-resistant cells, causing enhance in the transfer of antimicrobial resistance genes (ARGs) among pathogens [9]. As highlighted by separate reports, the production of carbapenemases efficiently prevents the activity of carbapenems and other β-lactams (except monobactams), and are often not inhibited by clinically available β-lactamase inhibitors, including clavulanic acid or tazobactam [10,11,12]. Additionally, bacteria co-expressing serine β-lactamases (SβLs) and metallo-β-lactamases (MβLs), two kinds of enzymes, are usually able to hydrolyze the clinically relevant monobactam, aztreonam [13].

The Ambler classification, which ranges from A to D, can be used to classify carbepenemases. Enzymes with serine in their active sites make up classes A, C, and D, whereas those with zinc in their active sites make up class B. The carbapenemases from *K. pneumoniae* (KPCs), imipenem-hydrolyzing β-lactamase (IMI), Guyana Extended Spectrumase Carbapenemase (GES), carbapenemase from *Serratia fonticola*, *Serratia marcescens* enzyme, and nonmetallo-carbapenemase-A are all examples of class A carbapenemases. Plasmids are where most of the genes for these class A enzymes are found. The genes encoding these enzymes can be found on plasmids, integrons, or even the bacterial chromosome itself in MβLs, which are class B organisms. Class C β-lactamases, like classes A and D, are based on serine and are encoded by genes that are often located on the bacterial chromosome. AmpC, a cephalosporinase, is the most important member of class C. Class D enzymes, sometimes referred to as oxacylinases (OXA), are typically produced by genes that are found on plasmids or the bacterial chromosome. Penicillins have excellent hydrolysis capabilities, whereas more recent generations of cephalosporins, such ceftazidime and cefepime, do not [14,15].

Among carbapenemases, the production of MβL has been associated with at least twelve (12) different types of MβLs, including Verona imipenemase—VIM, imipenemase—IMP, São Paulo MβL—SPM, German imipenemase—GIM, New Delhi MβL—NDM, Dutch imipemenase—DIM, Australian imipenemase—AIM, Seoul imipenemase—SIM, Kyorin University Hospital MβL—KHM, *Serratia marcescens* MβL—SMB, Tripoli MβL—TMB, and Florence imipenemase—FIM. Indeed, the mobile genetic element containing MβL in *P. aeruginosa* was originally reported in Japan at the end of the 1980s and has since been described in different parts of the world in association with several variants [16,17,18].

Regarding *bla*_SPM-1_, SPM-1 was initially thought to be contained in a plasmid, but further investigations on the gene for *bla*_SPM-1_ found that it was actually placed in movable genetic elements on the chromosome of the matching isolates of *P. aeruginosa* [19,20]. A broad spectrum MβL with a little bias for cephalosporins is SPM-1 [21]. The enzyme has an unusual structure since it lacks the conserved Asp84 and has a considerably shorter L3 loop. Instead, it has a 24-residue insertion in helix α3 that results in an extended helical segment, abutting the active region in a way reminiscent of B2 enzymes. Less than the deletion of the L3 loop in other B1 enzymes, the deletion of this extension of three reduces SPM-1’s catalytic activity by three- to thirty-two-fold [22,23].

Following a worldwide trend, MβL-producing *P. aeruginosa* isolates began to be described in Brazil in the early 2000s and are responsible for increased levels of resistance to carbapenems. They are similarly as observed in other locations, and CR-PA strains are often detected harboring *bla*_IMP_, *bla*_VIM,_ and *bla*_NDM_ variants in samples from various sources [24,25,26,27,28,29,30]. Contrastingly, the *bla*_SPM-1_ variant gene stands out for its endemic character and wide detection in different regions of the Brazilian territory, and from its first detection in 2002, it has been spread in the south region (in the State of Rio Grande do Sul—RS), the southeast region (in the State of São Paulo—SP), the northeast region (in the State of Sergipe—SE), the nidwest region, the State of Mato Grosso do Sul—MS, and the northern region (State of Pará) [19,31,32,33,34].

Despite the efforts in identifying and reporting cases of MβL-CR-PA in Brazil, a need for more robust studies is still noted in order to provide data that contemplate the large dimension of the Brazilian territory. Therefore, the present study aims to report, through a systematic review and meta-analysis support, the prevalence of MβL-CR-PA strains in Brazilian territory, as well as the prevalence of *bla*_SPM-1_ in comparison to other MβL gene variants. *bla*_SPM-1_, which is one of the MβL resistance genes distributed in gram-negative pathogens, was particularly chosen for the analysis in this article due to the fact that it is the most prevalent in Brazil and was originally isolated in SP [35].

## 2. Materials and Methods

### 2.1. Study Design

This is a systematic review and meta-analysis with the objective of investigating the aforementioned objective based on antimicrobial resistance data available in studies, including the detection of genetic mechanisms (MβLs) on *P. aeruginosa* isolated in the Brazilian territory. The present study followed the recommendations of the Preferred Report Items of a Systematic Review and Meta-Analyses (PRISMA) Statement 2020 [36]. This ratio-type meta-analysis was performed in this study for two correlated investigations: (I) to identify the prevalence of MβL-PA among Brazilian isolates and (II) to determine the prevalence of the *bla*_SPM-1_ variant in comparison to other MβL variants within the analyzed studies and isolates.

### 2.2. Search Strategy and Eligibility Criteria

To formulate the guiding question of the present systematic review, the POT strategy was used, and it consisted of the following question: “What is the prevalence of MβL-PA among Brazilian samples?” The anagram for its formation was composed of “P” for problem (MβL-CR-PA), “O” for outcome (frequency of MβL-PA in Brazil), and T for study type (original studies). The keywords used for the database search were Medical Subject Headings (MeSH): (((“beta-Lactamases/analysis” [Mesh]) OR (“beta-Lactamases/genetics” [Mesh])) AND “Brazil” [Mesh]) AND “*Pseudomonas aeruginosa*” [Mesh] [37].

The databases investigated for the search were PUBMED (https://pubmed.ncbi.nlm.nih.gov/ (accessed on 11 August 2023)), SciELO (https://www.scielo.br/ (accessed on 11 August 2023)), Science Direct (https://www.sciencedirect.com/ (accessed on 11 August 2023)), and LILACS (https://lilacs.bvsalud.org/ (accessed on 11 August 2023)), as those databases in the Latin American have weight and count most Brazilian publications. Primary studies published in Portuguese, English, or Spanish, within the time frame from January 2009 to August 2023, and of analytical case–control, cohort, and cross-sectional types, were eligible [38]. Brief/short communications, letters to the editor, editorials, articles unavailable in their complete form, articles outside the stipulated time cut, and the aforementioned inclusion criteria, were excluded.

### 2.3. Data Extraction and Methodological Quality Assessment

The data were collected in all databases on 11 August 2023. The database search, collection, investigation, tabulation, and extraction of the data were conducted by two authors independently (PASS and MJAS). Any disagreement between the analyses was resolved with help from a third researcher (YCR). The data extracted from the articles were authors, year of publication, title, sourced database(s), methodology, objective(s), sample type, Brazilian city/region, period in which the *P. aeruginosa* isolates were collected, MβL-PA frequency results, and if applicable, frequency of *bla*_SPM-1_ among MβL-PA.

The quality assessment was performed by two investigators (PASS and MJAS) using the Joanna Briggs Institute’s (JBI) Critical Appraisal Checklist for Analytical Cross-Sectional Studies (0–8), the JBI Checklist for Case–Control Studies (0–10), and the JBI Checklist for Cohort Studies (0–11) [39]. Only when the conditioned response was “Yes” were the scores for completing the checklist questions taken into account [40]. A third researcher (YCR) opined in case of disagreement.

### 2.4. Statistical Analysis

The Comprehensive Meta-Analyses—CMA program, version 2.2 (Biostat, Englewood, NJ, USA), was used on a computer to perform the statistical analyses of the meta-analysis for the two investigations. The fixed effects model estimated the combined frequency of *P. aeruginosa* with 95% confidence intervals (95% CI). A subgroup analysis was performed to determine the frequency of MβL-PA isolates according to Brazilian regions (northern—composed of seven states, northeast—composed of nine states, midwest—composed of three states, the Federal District, south—composed of four states, and southeast—composed of four states). The Cochrane Q-test and I-squared (I^2^) measure were used to determine the statistical difference groups (*p* < 0.05 was considered statistically significant). Begg’s rank correlation test and a funnel plot were used to examine the potential for publication bias (*p* < 0.05 was considered statistically significant). Sensitivity analysis, meta-regression, and subgroup analysis based on study location were used to assess potential causes of variability, where applicable [41].

## 3. Results

### 3.1. Literature Search

A total of 298 studies were found within the selected databases. After initial screening and application of eligibility and inclusion criteria, 173 were out of the scope proposed by this review, 24 studies were in languages other than those agreed upon in the methodology, 30 studies only had the abstract available, 14 studies were unavailable for full reading, 20 were duplicate studies, 1 was a letter to the editor, and 16 studies through full reading were excluded. The final sample and its stages until final inclusion were described in Figure 1.

### 3.2. Characterization of Included Studies

A total of 20 studies were included in the final analysis. All were presented in English, found on the PUBMED database, and had who were authors affiliated with Brazilian teaching and research institutions. Most of the studies were characterized as a cohort type (*n* = 14; 70%), with MβL-PA isolates recovered in a clinical/hospital setting (*n* = 17; 85%), followed by animal–environmental settings (15%) and a high detection frequency of the *bla*_SPM-1_ gene among isolates. The studies were evaluated by the JBI checklists considering each study design investigated and, in general, they showed high methodological quality (Table 1).

### 3.3. Results and Publication Bias of Meta-Analysis of Proportion of MβL-PA

In a generalized analysis of 20 studies, the prevalence rate of MβL-PA isolates was 35.8% or 0.358 (95% CI = 0.324–0.393). The studies’ differences were significantly different from one another (x^2^ = 243.15; *p* < 0.001; I^2^ = 92.18%), so they were divided into subgroups based on Brazilian geographic regions (Figure 2). This analysis demonstrated the northern region, with a frequency of 13.8% or 0.138 (95% CI = 0.080–0.227); the northeast region, with 26.4% or 0.264 (95% CI = 0.167–0.393); the midwest region (although only one study was included in the analysis), with 56.4% (95% CI = 0.450–0.671); the southeast region, with 28.9% or 0.289 (95% CI = 0.249–0.332); and finally the south region (although only two studies were included in the analysis), revealing the highest prevalence of MβL-producing P. aeruginosa, with 57.8% or 0.578 (95% CI = 0.499–0.653). A thematic map of these estimated percentual values in Brazilian regions was characterized in Figure 3. Despite the asymmetric distribution of the graph in the funnel plot (Figure 4), as it is a meta-analysis of the proportion type, the Begg test was performed. According to the findings of Begg’s test, no discernible publishing bias was found (z = 0.42; *p* = 0.67).

### 3.4. Meta-Regression of the Studies Included by Year of Publication

Meta-regression results by year of publication of the studies showed a significant association with heterogeneity in findings of the average prevalence rates of MβL-PA among Brazilian isolates in a fixed-effects model; coefficients −0.2574 (95% CI = −0.199–0.7138, *p* < 0.0001) (Table 2 and Figure 5).

### 3.5. Results and Publication Bias of the Meta-Analysis of bla_SPM-1_ within MβL-PA Isolates

In a generalized analysis of 19 studies, the prevalence rate of MβL-PA isolates in the presence of the bla_SPM-1_ was 66.9%, ranging from 59.3% to 73.8%, with a 95% CI (Figure 6). Only one study was excluded in this analysis, as only one MβL-PA isolate was found, and the sample size for comparison has to be higher than or equal to two [48]. The heterogeneity of the results among the included studies was considered average (x^2^
*=* 90.93; *p* < 0.001; I^2^
*=* 80.20%). The distribution of the studies in the graph in Figure 6 was symmetrical, and Begg’s test reported non-significance (z *=* 0.48; *p =* 0.62), demonstrating no publication bias for this meta-analysis (Figure 7).

### 3.6. Comparison between the Results of this Brazilian Meta-Analysis and Prevalence Worldwide

Previous reviews have already reported the prevalence of MβL-PA in hospital isolates globally, especially in Asia. Ghasemian et al. (2018) in Iran composed a review of 36 studies in 2018 and reported an incidence of 37.72% MβL-PA, ranging from 16.68% to 100%, with no reports of isolates harboring *bla*_NDM-1_ and *bla*_SPM-1_ [58]. Another meta-analysis by Dawadi et al. (2022), conducted in Nepal, found a frequency of 14% MβL-PA (95% CI *=* 0.10–0.19) for data pooled from studies published until 2021 in Nepal, and MDR-PA isolates were about 42% in Nepal [59] (Figure 8).

## 4. Discussion

Although they are known as “carbapenemases”, many of these enzymes recognize nearly all hydrolyzable β-lactams, and most are resistant to inhibition by all commercially viable β-lactamase inhibitors. With the exception of the SPM type enzyme, most MβL genes reside in various integron cassette compositions attached to mobile elements, which is a condition that facilitates their dissemination among different bacterial species through horizontal gene transfer [60,61]. This scenario is in association with the poor consequences of unrestricted antimicrobial usage and heterogeny in the social–economic context worldwide. CR-PA has proved to be a public and healthcare emergency [62,63,64,65,66]. Hence, the present study aims to provide robust data on the prevalence of MβL-PA isolates in Brazil, as well as the prevalence of the endemic *bla*_SPM-1_ resistance gene.

In the global antimicrobial resistance epidemiological scenario, CR-PA has been reported to contain a wide variety of carbapenemases. In Latin America, the Arabian Peninsula, and the United States of America, this includes KPC, GES, IMP, VIM, NDM, and SPM [67,68,69,70]. Approximately 49.4% of CR-PA from Spain tested positive for MβL, in which the *bla*_VIM-2_ was detected in almost all isolates. In addition, 75% tested positive for integron class 1 [71]. Class 1 and 2 integrons were found among 60% of the carbapenem-resistant isolates in Colombia that were VIM-positive but were all IMP- and NDM-negative [72]. Germany, Switzerland, Brazil, and Italy reported GIM-1-, SPM-1-, and FIM-1-producing strains, respectively [73,74,75,76]. The predominant MβL-encoding gene in the Middle East and the majority of other Asian nations is the *bla*_VIM_ and its variants. Recently, the emergence of the NDM-1-producing *P. aeruginosa* is quite concerning because it first appeared in nations other than India [77].

Brazil also has proven to be an important ally in the detection of MDR/XDR CR-MβL-PA strains, as highlighted by data from the present study, in which IMP-, VIM-, and SPM-producing isolates have been mostly detected [17,78,79,80]. Worryingly, there is a recent report of CR-PA harboring the *bla*_NDM-1_ gene along with other MβL [29]. Indeed, the occurrence of isolates with the co-production of MBLs has been verified, as reported in some studies included in this meta-analysis [31,47]. As opposed to to other MβL found in lower frequency, the endemic feature is key to the *bla*_SPM-1_ gene, as there are scarce reposts outside Brazil, Europe [73], Iran [81,82], the UK [83], Chile [84], and Egypt [85]. This fact must be related to the clonal dissemination of MβL-PA in association outbreaks by the ST277, as described in several studies in different Brazilian regions, and the chromosomal nature of this variant, which may hinder its dissemination [42,50,56,86].

This is a pioneer review and meta-analysis study involving the prevalence of MβL-PA isolates in Brazil and providing significant data on the endemic *bla*_SPM-1_ gene frequency. Previous data from the two studies cited in Section 3.6 of this article highlight a similar profile of results presented in this meta-analysis based on Brazilian territory isolates. This is an emerging problem worldwide [58,59].

Other meta-analyses present in the literature, such as those conducted by Ul Ain et al. (2020) [87] and Osei Sekyere et al. (2021) [88], performed a broader and less specific analysis of *P. aeruginosa* in Pakistan by compiling a brief description and, for the first time, a report of the resistant carbepenemases rate (RC) for different gram-negative bacteria, including *P. aeruginosa*. In the first study, the prevalence of MβLs from different geographical regions of Pakistan showed concerning high rates of RCs and incidences of MβLs; while the second evaluated the global distribution of the rate of carbapenem resistance in gram-negative bacterial strains in pregnant hosts and children in comparison with polymyxin resistance and death rate in the countries, in which Brazil had a CR rate of approximately 54.5% and death rate higher than 10%. Vaez et al. (2018), in a meta-analysis in Iran in the period up to 2018, determined a prevalence of about 32.4% of MDR-PA, and the most reported MβL genes were *bla*_VIM_ and *bla*_IMP_, with frequencies of 19% (95% CI: 0.15–0.23) and 11% (95% CI: 0.08–0.14) [89], respectively.

Nationally, only one meta-analysis published by our research group includes isolates from the Brazilian population, which under a time cut between 2006 and 2016, investigated the mortality rate related to MDR-PA and SPM-1-producing strains. The results demonstrated a higher mortality rate among MDR-PA-infected patients (44.6%, 363/813) than those with a non-MDR-PA infection (24.8%, 593/2. 388) (OR = 2.39, 95% CI = 1.70–3.36, *p* < 0.000017) [7].

Our study demonstrated that most of the samples are from clinical/hospital settings, in which CR-PA is one of the main pathogens involved in infections, colonization, and invading things like respiratory treatment machines, disinfectants, sinks, distilled water, central venous catheters, urine catheters, and even the hands of medical staff [90]. Humans also expel resistant bacteria into the effluent, which may spread to other environmental compartments. The selection of resistant bacteria, such as those related to some Brazilian regions of this study, is influenced by the selective pressure applied by antibiotics often found in hospital effluent [91].

Depending on the type of geographic region and the type of methodology applied in the primary study, the results tend to be different. This is due to the socioeconomic, environmental, and zooanthropological characteristics of the investigated region, in addition to the type of sample analyzed, such as the prevalence of this etiological agent in surgical and respiratory wounds or in urinary and surgical infections (where there is usually a higher occurrence) [92]. In this case, this meta-analysis shows that there is a large number of studies covering the southeast region that may come from the suggestion of several factors that still need to be better investigated together, such as the hospital environments themselves and the outbreaks of this pathogen, which present a relative frequency of MβL-PA that still constitutes a relevant public health problem. The south region of Brazil has been identified as the region with the highest incidence rate of MΒL-producing strains. In contrast, numerous other evaluated studies have indicated that the rate of MβL-*P. aeruginosa* varies. In the northern region, our research group conducted molecular epidemiology studies providing significant data on SPM-1 clinical impact and the role of high-risk clone ST277 on the spread of this variant in the region [32,49]. This heterogeneity may be related to several elements, including antibiotic availability and use, variations between nosocomial surveys, and techniques to identify MβL producers [59].

The limitations of this present systematic review and meta-analysis conducted in Brazil lie in terms of the search methodology applied (due to the use of specific keywords for the topic) and the inclusion of studies focusing on evaluating MβL detection by molecular methods, even though some include data on phenotypical detection. Also, in this meta-analysis, only original articles were included; however, it was observed that there are articles, mainly in a short communication format, which were excluded, which reported a high frequency of MβL and SPM-1 gene-producing strains in the Brazilian regions, which demonstrates that the estimated prevalence of MβL and SPM-1 may still be underestimated in their epidemiological natures [44,93,94,95]. Even though other sources may have been excluded from this review due to their in-depth profile or lack of high-quality evidence, this article denotes a good standard of methodological and scientific evidence and provides a basis for one-health strategies and new studies on this topic.

## 5. Conclusions

MβL production by *P. aeruginosa* has been proven as complicating factor, not only for the worsening of infections but also for the successful treatment of infected patients, thus making it a public health problem, especially in underdeveloped and emerging countries, such as Brazil, where hospitals lack better structures, the health system is defective, and available resources are limited, especially for the poor population. According to our research results, the prevalence of MβL-PA is alarmingly high in most Brazilian hospitals and clinical settings, along with the detection in animal–environmental interfaces, similar to other places in the world, such as Iran, making it necessary to develop effective dissemination control strategies. It was also possible to verify, through data analysis, that the high prevalence of the resistance gene *bla*_SPM-1_ compares with the high dissemination rate of other MΒL genes already detected in Brazil. Finally, we highlight the need for new studies for the better characterization of CR-MβL-PA in the distinct Brazilian regions with a focus on molecular epidemiology-based studies supported by new genomics approaches and technology, which aim to investigate outbreaks in endemic regions and the presence of such strains in non-clinical settings. New insights into the resistome features may be provided.

## Figures and Tables

**Figure 1 microorganisms-11-02366-f001:**
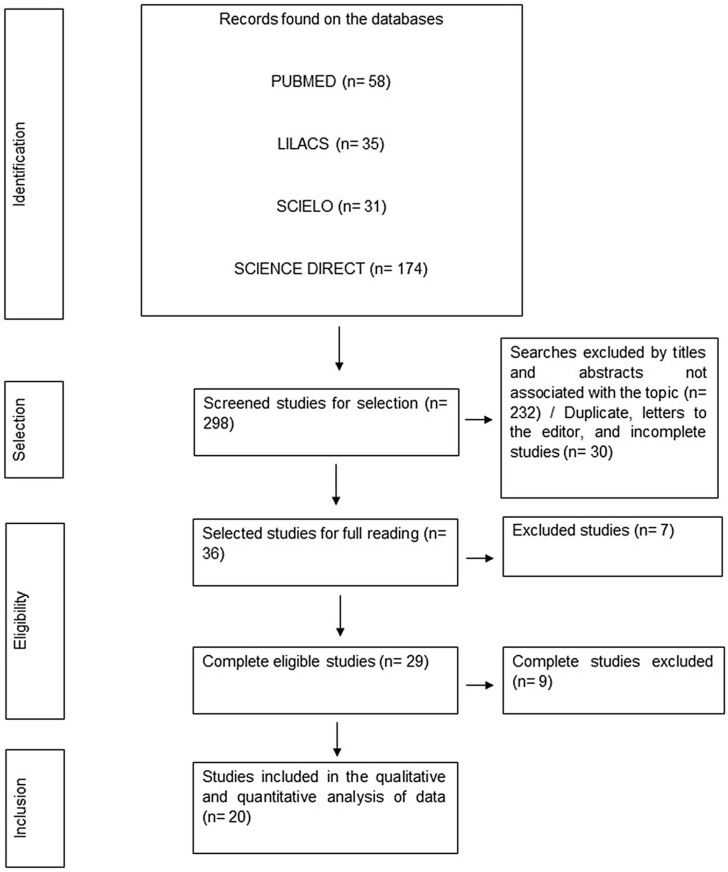
Flowchart of data collection and screening stages. Belém-PA, Brazil (2023).

**Figure 2 microorganisms-11-02366-f002:**
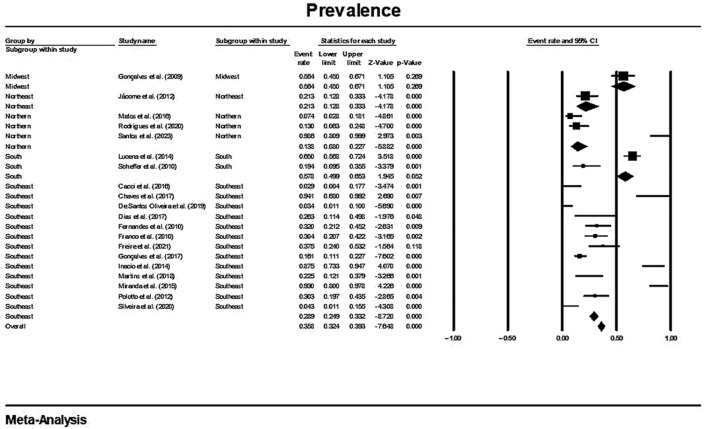
Forest plot of the prevalence of MβL-PA isolates by sampling PA isolates grouped by Brazilian regional subgroups. The size of each square, which corresponds to the weight of the related research in the meta-analysis, represents the OR of each study on the map. The 95% confidence intervals (CIs) for each study’s OR are shown as horizontal lines. The bold figures indicate the overall OR and the 95% CI, as well as the total frequency of cases and controls [31,32,33,34,42,43,44,45,46,47,48,49,50,51,52,53,54,55,56,57].

**Figure 3 microorganisms-11-02366-f003:**
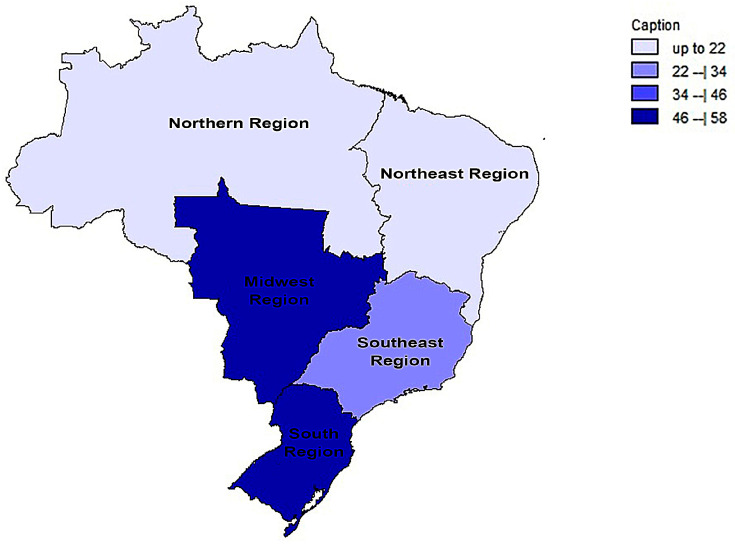
Thematic map of the estimated percentual value (%) of MβL-producing *P. aeruginosa* strains in Brazilian national distribution over the years of this review based on official regional groupings.

**Figure 4 microorganisms-11-02366-f004:**
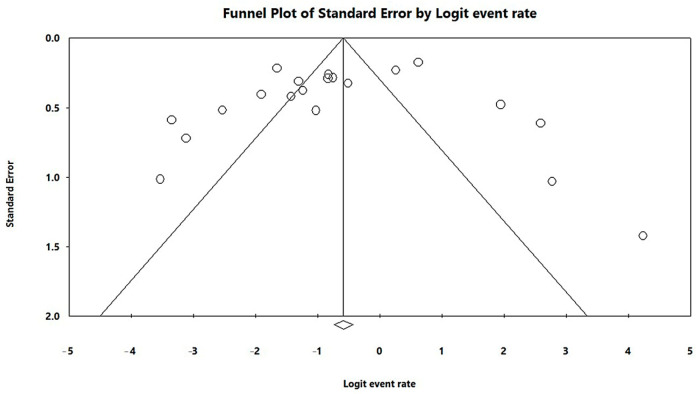
Funnel plot of the heterogeneity assessment among the studies included in the meta-analysis. The included published studies are represented by circles, which should be evenly spaced around the overall effect to resemble an inverted funnel. Studies that are more precise are in the narrowest region of the funnel and are closest to the real value. The standard error, which is represented on the graph’s Y-axis as a measure of dispersion, is affected by the study’s sample size. The bigger this value, the more inaccurate the study is. The center line of the graph, which is pointed at the X-axis by the diamond, represents the outcome of the effect measure examined in the meta-analysis. The lines that make up the funnel’s outline line up with 95% CI.

**Figure 5 microorganisms-11-02366-f005:**
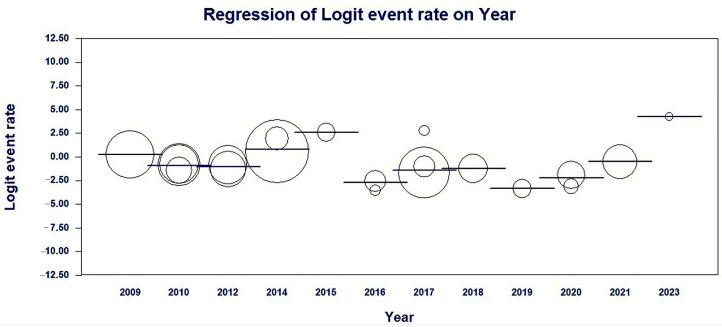
Scatterplot of the meta-regression of the logit event rate (MβL-PA/PA) by study publication year covariate. The size of the bubble is inversely related to the variance of the study. The solid line represents the linear regression (continent as the meta-independent variable).

**Figure 6 microorganisms-11-02366-f006:**
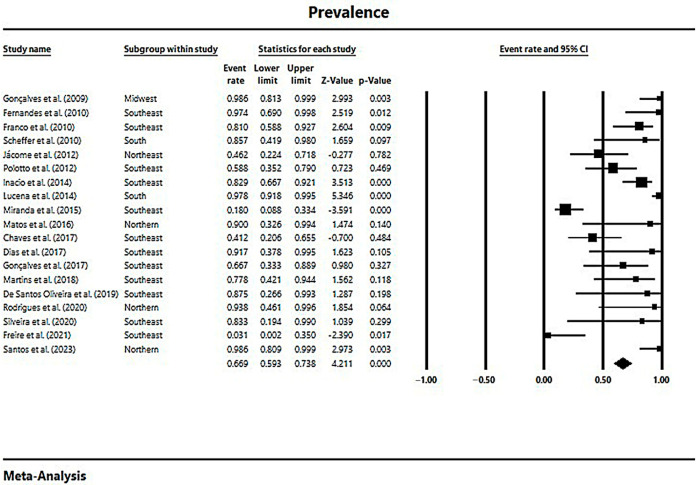
Forest plot of the prevalence of *bla*_SPM-1_ among MβL-PA isolates in the Brazilian territory. The size of each square, which corresponds to the weight of the related research in the meta-analysis, represents the OR of each study on the map. The 95% confidence intervals (CIs) for each study’s OR are shown as horizontal lines. The bold figures indicate the overall OR and the 95% CI, as well as the total frequency of cases and controls [31,32,33,34,42,43,44,45,46,47,49,50,51,52,53,54,55,56,57].

**Figure 7 microorganisms-11-02366-f007:**
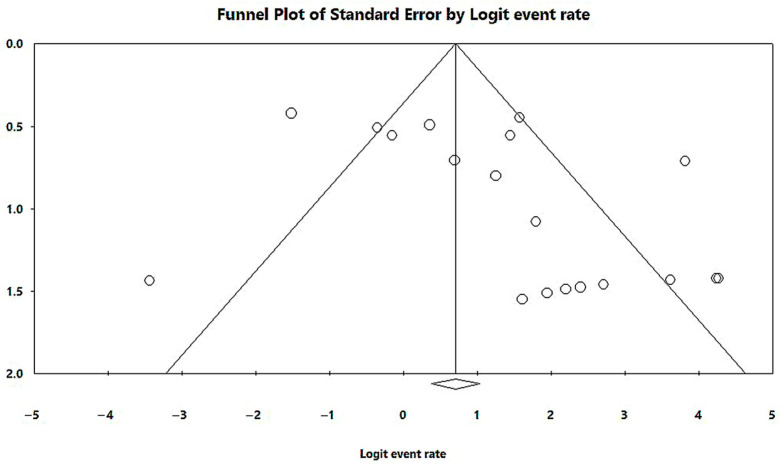
Funnel plot of the heterogeneity assessment among the studies included for the analysis of the *bla*_SPM-1_. The included published studies are represented by circles, which should be evenly spaced around the overall effect to resemble an inverted funnel. Studies that are more precise are in the narrowest region of the funnel and are closest to the real value. The standard error, which is rep-resented on the graph’s Y-axis as a measure of dispersion, is affected by the study’s sample size. The bigger this value, the more inaccurate the study is. The center line of the graph, which is pointed at the X-axis by the diamond, represents the outcome of the effect measure examined in the meta-analysis. The lines that make up the funnel’s outline line up with 95% CI.

**Figure 8 microorganisms-11-02366-f008:**
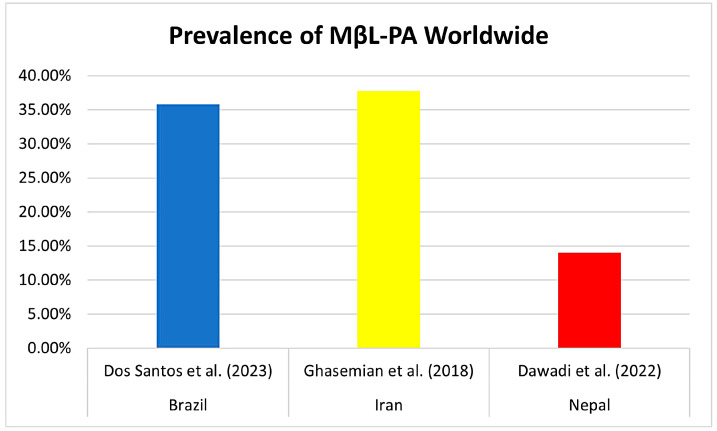
Comparison of the prevalence (%) of MβL-PA isolates already reported in the literature in countries around the world. The first study by Dos Santos et al. (2023) cited in the graphic is about the results of this meta-analysis [58,59].

**Table 1 microorganisms-11-02366-t001:** Characteristics of the studies included in this systematic review.

N^o^	Authors and Year of Publication	Title	Database	Methodology/Sample Size	Purpose(s)/Objective(s)	Sample Type/City (Region)/Period	Results	JBI Score	Frequency of MβL Resistance Genes among *P. aeruginosa* Isolates
1	Gonçalves et al. (2009) [42]	Detection of metallo-beta-lactamase in *Pseudomonas aeruginosa* isolated from hospitalized patients in Goiânia, State of Goiás	LILACS and SciELO and PUBMED	Cohort/75 isolates from patients with PA infection.	To investigate the susceptibility profile of *P. aeruginosa* previously isolated from patients in a hospital in Goiânia (GO).	Clinical/hospital settings/Goiânia-GO (midwest)/2005–2007	Among the 62 isolates resistant to IMP and CAZ, 35 (56.4%) produced MβL, corresponding to 47.3% of the PA isolates, while 26 (74.3%) presented the *bla*_SPM-1_ gene.	(11/11)	The *bla*_SPM-1_ gene was detected in high frequency (26/35–74.28%).
2	Fernandes et al. (2010) [43]	Molecular characterization of carbapenem-resistant and MβL-producing *Pseudomonas aeruginosa* isolated from blood cultures from children and teenagers with cancer.	LILACS and SciELO and PUBMED	Cohort/56 PA isolates from 49 patients.	Assess the persistence of MβL-PA from BSIs.	Clinical/hospital settings/São Paulo-SP (southeast)/2000–2005	A total of 32 (57.1%) CR-PA and, of this amount, 18 samples (32.14%) were positive for MΒL by a disk approximation test, equivalent to 17.85% of the total. All 18 samples were positive for the *bla*_SPM-1_ gene. The *bla_IMP-1_*, *bla_VIM-1_*, *bla_VIM-2_* were not detected.	(11/11)	The *bla*_SPM-1_ gene was related to all CR-PA (18/18–100%).
3	Franco et al. (2010) [33]	Metallo-beta-lactamases among imipenem-resistant *Pseudomonas aeruginosa* in a Brazilian university hospital	LILACS and SciELO and PUBMED	Cohort/69 included PA isolates.	Determine the frequency of MβL-PA among imipenem-resistant isolates in this hospital and compare phenotypic and molecular methods of detection.	Clinical/hospital settings/São Paulo-SP (southeast)/2006	The frequency of MβL-PA strains was 30.4% (*n* = 21). Of that number, 17 (80.95%) were positive for *bla*_SPM-1_ and 4 (19.05%) for *bla_VIM-2_*. The *bla_IMP-1_*, *bla_IMP-2_,* and *bla_VIM-1_* were not detected.	(10/11)	Distinct variants were detected, with a higher frequency of the *bla*_SPM-1_ (17/21–80.95%), followed by *bla_VIM-2_* (4–19.05%).
4	Scheffer et al. (2010) [44]	Intrahospital spread of carbapenem-resistant *Pseudomonas aeruginosa* in a University Hospital in Florianópolis, Santa Catarina, Brazil	LILACS and SciELO and PUBMED	Cohort/36 CR-PA isolates.	To investigate the minimum inhibitory concentration (MIC), the presence of MβL-PA and a possible clonal relationship between isolates in one teaching hospital in southern Brazil.	Clinical/hospital settings/Florianópolis–SC (south)/2003–2005	Seven isolates (19.44%) presented positive phenotypic tests for MβL, and most carried the *bla*_SPM-1_ gene.	(11/11)	Distinct variants were detected, with a higher frequency of the *bla*_SPM-1_, (6/7–85.71%) and *bla_IMP-16_* (1–14.29%).
5	Jácome et al. (2012) [34]	Phenotypic and molecular characterization of antimicrobial resistance and virulence factors in *Pseudomonas aeruginosa* clinical isolates from Recife, State of Pernambuco, Brazil	LILACS and SciELO and PUBMED	Cohort/61 PA strains from public hospitals.	Identify resistance and virulence markers and establish clonal spread of PA isolates.	Clinical/hospital settings/Recife-PE (northeast)/2006–2010	A total of 29 isolates were resistant to imipenem and/or ceftazidime. From this total, 44.8% (13/29) were MβL-PA, of which, 46.2% (6/13) revealed the *bla*_SPM-1_ gene.	(9/11)	The *bla*_SPM-1_ gene was related to 13 (46.15%) MβL-PA.
6	Polotto et al. (2012) [45]	Detection of *P. aeruginosa* harboring *bla* CTX-M-2, *bla* GES-1 and *bla* GES-5, *bla* IMP-1 and *bla* SPM-1 causing infections in Brazilian tertiary-care hospital	PUBMED	Cohort/56 *P. aeruginosa* isolates.	To investigate the carriage of genes codifying MΒLs and ESBLs by *P. aeruginosa* isolated from patients admitted to a Brazilian 720-bed teaching tertiary care hospital.	Clinical/hospital settings/São Paulo—SP (Sudeste)/June–December 2009	The prevalence of isolates harboring MβL genes was 30.3% (17/56). Ten (10/17 58.82%) presented the *bla*_SPM-1_ gene, and the *bla_IMP-1_* was detected in seven isolates (41.17). No *bla_VIM_* type was detected.	(10/11)	Distinct variants were detected, with a higher frequency of the *bla*_SPM-1_ gene.
7	Inacio et al. (2014) [31]	Phenotypic and genotypic diversity of multidrug-resistant *Pseudomonas aeruginosa* Isolates from bloodstream infections recovered in the Hospitals of Belo Horizonte, Brazil	PUBMED	Cohort/40 PA isolates from 5 hospitals in the city of Belo Horizonte.	To evaluate antimicrobial resistance and the production of MβL, oxacillinase, and cephalosporinases, and the genetic diversity of *P. aeruginosa* strains isolated from patients with BSIs.	Clinical/hospital settings/Belo Horizonte-MG (southeast)/2008–2009	A frequency of 87.5% (*n* = 35/40 were MβL-producers. Of this amount, 29 isolates were exclusively detected harboring the *bla_SPM-1_* gene (82.86%), equivalent to 72.5% of the total PA samples. Simultaneous occurrence of the *bla*_SPM-1_ and *bla_VIM-1_* genes was observed in six MβL-PA.	(10/11)	High frequency of the *bla*_SPM-1_ (82.86%) gene. Strains co-harboring the *bla*_SPM-1_ and *bla_VIM-1_* genes (17.14%).
8	Lucena et al. (2014) [46]	Nosocomial infections with metallo-beta-lactamase-producing *Pseudomonas aeruginosa*: molecular epidemiology, risk factors, clinical features and outcomes	Science Direct and PUBMED	Cohort/CR-PA were isolated from different clinical samples of hospitalized patients.	To evaluate the molecular epidemiology, risk factors, and outcomes of nosocomial infections caused by MβL-PA in a university hospital in southern Brazil.	clinical/hospital settings/Curitiba-PR (south)/2001–2008	A total of 93/142 (65%) strains were confirmed as MβL-PA. Considering the total number of isolates, 91 isolates contained the *bla*_SPM-1_ gene (64%), followed by the *bla_IMP-1_* (0.7%) and the *bla_IMP-16_* (0.7%) genes in one isolate each.	(10/10)	Three distinct variants were detected among MβL-PA, with emphasis on the *bla*_SPM-1_ gene (97.84%), *bla_IMP-1_* (1.08%), *bla_IMP-16_* (1.08%).
9	Miranda et al. (2015) [47]	Genotypic characteristics of multidrug-resistant *Pseudomonas aeruginosa* from hospital wastewater treatment plant in Rio de Janeiro, Brazil	PUBMED	Cohort/41 isolates of PA.	To investigate the antimicrobial susceptibility profiles and genetic relatedness of *P. aeruginosa* isolates obtained from a Hospital Wastewater Treatment Plant (HWTP).	Environmental/Rio de Janeiro—RJ (southeast)/2008–2010	A total of 38 (93%) isolates exhibited antimicrobial resistance to carbapenems.	(11/11)	Among these, 14 (37%) were *bla_VIM_* and six were (18%) *bla*_SPM- 1_-positive. Three strains were found co-harboring the *bla*_VIM_ and *bla*_SPM-1_ genes.
10	Cacci et al. (2016) [48]	Mechanisms of carbapenem resistance in endemic *Pseudomonas aeruginosa* isolates after an SPM-1 metallo-β—lactamase producing strain subsided in an intensive care unit of a teaching hospital in Brazil	LILACS and SciELO and PUBMED	Cohort/35 clinical and 1 hemodialysis isolate of PA from 234 patients admitted to the ICU.	Compare the resistance levels of clinical isolates of *P. aeruginosa* detected from 2007 to 2008 with those previously described in the literature.	Clinical/hospital settings/Rio de Janeiro-RJ (southeast)/2007–2008	CR-PA was related to one *bla*_SPM-1_ positive isolate. The *bla*IMP-type, *bla*VIM-type, *bla*SIM-1, *bla*GIM-1, and *bla*NDM-1 were not detected.	(10/11)	One *bla*_SPM-1_ (100%) in MβL-PA
11	Matos et al. (2016) [49]	Clinical and microbiological features of infections caused by *Pseudomonas aeruginosa* in patients hospitalized in intensive care units	LILACS and SciELO and PUBMED	Cohort study/54 isolates from patients in beds admitted to intensive care units (ICUs).	To analyze the clinical, epidemiological, and molecular characteristics of *P. aeruginosa* infections in ICU patients at a university hospital.	Clinical/hospital settings/Belém -PA (northern)/2010–2012	The frequency of MDR-PA strains was 37% (20/54). Of this amount, 20% (4/20) of the isolates were positive for the MβL *bla*_SPM-1_ gene, which is equivalent to 7.4% of the total samples.	(11/11)	The *bla*_SPM-1_ gene was found in four MβL isolates.
12	Chaves et al. (2017) [50]	High mortality of bloodstream infection outbreak caused by carbapenem-resistant *P. aeruginosa* producing SPM-1 in a bone marrow transplant unit	PUBMED	Case–control/29 cases and 58 controls, with 17 isolates.	To analyze the mortality and risk factors for developing BSI caused by carbapenem-resistant *P. aeruginosa* (CRPA).	Clinical/hospital settings/São Paulo—SP (southeast)/2011–2013	Most isolates were MβL-PA (16/17—94.12%).	(10/11)	Seven isolates were found harboring the *bla*_SPM-1_ (41.2%)
13	Dias et al. (2017) [51]	Epidemiological, Physiological, and Molecular Characteristics of a Brazilian Collection of Carbapenem-Resistant *Acinetobacter baumannii* and *Pseudomonas aeruginosa*	PUBMED	Cross-sectional/19 PA were isolated from a private hospital.	To determine antimicrobial susceptibility and biocide tolerance patterns, hemolytic activity, biofilm formation, and oxidative stress tolerance in carbapenem-resistant and drug-sensitive bacteria, and also track genetic markers of carbapenem resistance of *A. baumanni* and *P. aeruginosa.*	Clinical/hospital settings/Juiz de Fora-MG (southeast)/January to December 2013	About five samples showed the MβL phenotype with the *bla*_SPM-1_ genetic marker (26.32% of the total). The *bla*IMP, *bla*VIM, *bla*SIM, *bla*GIM, and *bla*NDM-1 were not detected.	(8/8)	The *bla*_SPM-1_ gene was found in all 5 MβL-PA samples.
14	Gonçalves et al. (2017) [52]	Carbapenem-resistant *Pseudomonas aeruginosa*: association with virulence genes and biofilm formation	LILACS and SciELO and PUBMED	Cohort/56 CR-PA isolates from patients.	Identify strains of *P. aeruginosa* resistant to carbapenems.	Clinical/hospital settings/Uberlândia -MG (southeast)/2009–2012	Of the total CR-PA isolates, nine were phenotypically positive for MβL.	(10/11)	Among the MβL-PA, six (66.67%) were positive for the *bla*_SPM-1_ and three (33.33%) for the *bla_VIM_* genes.
15	Martins et al. (2018) [53]	SPM-1-producing *Pseudomonas aeruginosa* ST277 clone recovered from microbiota of migratory birds	PUBMED	Cohort/40 strains of *P. aeruginosa*.	To report the detection and molecular characterization of the SPM-1-producing *P. aeruginosa* ST277 clone recovered from the microbiota of migratory birds (Dendrocygna viduata) in Brazil.	Animal-related/São Paulo—SP (southeast)/July–August 2012	A total of nine CR-PA isolates (22.5%) were MβL producers.	(11/11)	The *bla*_SPM-1_ gene was found in 7 strains (77.77%).
16	De Oliveira Santos et al. (2019) [54]	Epidemiology and antibiotic resistance trends in clinical isolates of *Pseudomonas aeruginosa* from Rio de janeiro—Brazil: Importance of mutational mechanisms over the years (1995–2015)	PUBMED and Science Direct	Cohort/88 isolates of PA.	To determine patterns and mechanisms of antimicrobial resistance and its spread over the years in Rio de Janeiro.	Clinical/hospitalar/Rio de Janeiro—RJ (southeast)/2006–2010	A total of three *P. aeruginosa* strains were MβL producers (3.4%).	(10/11)	All three MβL-PA isolates were *bla*_SPM-1_ (100%)
17	Rodrigues et al. (2020) [32]	High prevalence of atypical virulotype and genetically diverse background among *Pseudomonas aeruginosa* isolates from a referral hospital in the Brazilian Amazon	PUBMED	Cross-sectional study/54 PA isolates from an Amazonian referral hospital.	To report in-depth data on virulence, resistance properties, and genetic diversity of *P. aeruginosa* isolates from patients admitted to the ICU of a reference hospital in the Brazilian Amazon region.	Clinical/hospitalar/Belém -PA (northern)/2010–2013	A *bla*_SPM-1_ frequency of 9.2% (*n* = 5/54) was found. Two isolates co-harbored the *bla*_SPM-1_*/bla_OXA-2_* and *bla*_SPM-1_*/bla_OXA-10_* genes, and one isolate was positive for *bla*_SPM-1_ only. The *bla*IMP, *bla*VIM, *bla*NDM, and *bla*KPC genes were not detected.	(8/8)	The *bla*_SPM-1_ was found in five MβL-PA.
18	Silveira et al. (2020) [55]	Exploring the success of Brazilian endemic clone *Pseudomonas aeruginosa* ST277 and its association with the CRISPR-Cas system type I-C	PUBMED	Cohort/47 strains of *P. aeruginosa*.	To examine the phylogenetic distribution and conservation of genetic determinants among ST277 *P. aeruginosa* genomes available at NCBI.	Environmental/Rio de Janeiro—RJ (southeast)/1997–2018	A total of two SPM-1 isolates were recovered from impacted rivers (4.25%).	(9/11)	All two samples were 100% *bla*_SPM-1_.
19	Freire et al. (2021) [56]	Critical points and potential pitfalls of outbreak of IMP-1-producing carbapenem-resistant *Pseudomonas aeruginosa* among kidney transplant recipients: a case control study	PUBMED	Case–control/40 CRPA isolates.	To analyze an outbreak of infection/colonization with IMP-1-producing CRPA on a kidney transplantation (KT) ward.	Clinical/hospitalar/São Paulo—SP (southeast)/2019–2020	A total of 15 *P. aeruginosa* strains were MβL producers (37.5%).	(10/11)	The *bla*_SPM-1_ gene was not found. All 15 strains were *bla*_IMP-1_ (100%).
20	Dos Santos et al. (2023) [57]	Endemic high-risk clone ST277 is related to the spread of SPM-1-producing *Pseudomonas aeruginosa* during the COVID-19 pandemic period in the Brazilian Northern region	PUBMED	Cross-sectional study/34 PA isolates.	To investigate the antimicrobial resistance, virulence, and genotypic characteristics of SPM-1-producing *P. aeruginosa* recovered from the pre-pandemic era in medical facilities in the states of Pará (PA) and Acre (AC), northern Brazilian Amazon area.	Clinical/hospitalar/Pará State (northern region) 2018–2022	All samples were classified as CRPA and MβL producers (*n* = 34; 100%).	(8/8)	All samples were 100% *bla*_SPM-1_ (100%).

**Table 2 microorganisms-11-02366-t002:** Main results for model 1, fixed effect, Z-distribution, and logit event rate.

Set	Covariate	Coefficient	Standard Error	95% Lower	95% Upper	Z-Value	2-Sided *p*-Value	Set
	Intercept	0.2574	0.2329	−0.199	0.7138	1.11	0.269	
Year	2010	−1.1612	0.2917	−1.7329	−0.5895	−3.98	0.0001	Q = 212.13, df = 11, *p* < 0.0001
Year	2012	−1.31	0.3155	−1.9284	−0.6915	−4.15	0	Q = 212.13, df = 11, *p* < 0.0001
Year	2014	0.5199	0.2855	−0.0396	1.0794	1.82	0.0686	Q = 212.13, df = 11, *p* < 0.0001
Year	2015	2.3293	0.6549	1.0457	3.6128	3.56	0.0004	Q = 212.13, df = 11, *p* < 0.0001
Year	2016	−2.9922	0.5181	−4.0076	−1.9768	−5.78	0	Q = 212.13, df = 11, *p* < 0.0001
Year	2017	−1.6585	0.3048	−2.256	−1.061	−5.44	0	Q = 212.13, df = 11, *p* < 0.0001
Year	2018	−1.4942	0.4445	−2.3654	−0.6229	−3.36	0.0008	Q = 212.13, df = 11, *p* < 0.0001
Year	2019	−3.6042	0.6326	−4.8441	−2.3643	−5.7	0	Q = 212.13, df = 11, *p* < 0.0001
Year	2020	−2.4511	0.4233	−3.2807	−1.6215	−5.79	0	Q = 212.13, df = 11, *p* < 0.0001
Year	2021	−0.7682	0.4011	−1.5544	0.0179	−1.92	0.0555	Q = 212.13, df = 11, *p* < 0.0001
Year	2023	3.9767	1.4433	1.1478	6.8056	2.76	0.0059	Q = 212.13, df = 11, *p* < 0.0001

## Data Availability

The original contributions of the study are included in the article. Further inquiries can be directed to the corresponding authors.

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
