# Peer review of "The Prevalence of Metallo-Beta-Lactamese-(MβL)-Producing Pseudomonas aeruginosa Isolates in Brazil: A Systematic Review and Meta-Analysis"

_microorganisms, 2023, doi:10.3390/microorganisms11092366_

Round 1

Reviewer 1 Report

Report

The present review article microorganisms-2552869 titled: “Prevalence of metallo-beta-lactamese-(MβL)-producing Pseudomonas aeruginosa isolates in Brazil: A systematic review and meta-analysiswas aimed at determining the prevalence of Pseudomonas aeruginosa (PA) producing MΒL among Brazilian samples and the frequency of blaSPM-1 in MΒL-PA producing isolates. From 2009 to March 2023: The topic is very interesting from the medical and public health aspects, the English language is very well and the study-design, is well presented; however, there are some major and minor comments and suggestions that should be considered and fulfilled, and these are as follows:

1.     In the abstract, the author should define the aim and the rationale of conducting this meta-analysis, particularly for focusing on blaSPM, which is one of the metallo-beta lactamases distributed in Gram-Negative pathogens. 2.     In the introduction, L55, I recommend changing “between bacteria” to “among pathogens). 3.     In the introduction, I recommend including some information about carbapenemases other than MBL and the criteria of each one as well as more elaborative information about blaSPM and whether it is plasmid-meditated or not with citation of relevant literature. 4.      L102-103, the authors should provide the URl of each online website together with the accession date for each one. 5.     L56, 57, the author should add. (It is worth mentioning that bacteria co-expressing SBLs and MBLs are usually able to hydrolyze the clinically relevant monobactam, aztreonam with citation of the relevant literature such as. https://doi.org/10.2144/fsoa-2019-0098; https://doi.org/10.1016/j.tim.2011.09.005 6.     The abbreviations should be mentioned in full at first mention, the whole manuscript should be accordingly revised. 7.     P6, what is the rationale for the yellow highlighting of  Santos et al. 8.     L168, P. aeruginosa should be italicized. The whole manuscript should be revised accordingly. 9.     Figures 2-7, I recommend including better-resolution ones. 10.  I recommend including a section in the results for comparing the prevalence of MBL in Brazil to that of other geographical areas worldwide particularly those of similar standards of living and in order to have a picture of the obtained percentage in relation to other areas.

Therefore, and for the above-mentioned remarks, I advised a minor revision of the respective manuscript in its current state taking into consideration the above comments and recommendations before being considered for publication.

Author Response

Report – Reviewer #1

 The present review article microorganisms-2552869 titled: “Prevalence of metallo-beta- lactamese-(MβL)-producing Pseudomonas aeruginosa isolates in Brazil: A systematic review and meta-analysiswas aimed at determining the prevalence of Pseudomonas aeruginosa (PA) producing MΒL among Brazilian samples and the frequency of blaSPM-1 in MΒL-PA producing isolates. From 2009 to March 2023: The topic is very interesting from the medical and public health aspects, the English language is very well and the study-design, is well presented; however, there are some major and minor comments and suggestions that should be considered and fulfilled, and these are as follows:

  1. In the abstract, the author should define the aim and the rationale of conducting this meta-analysis, particularly for focusing on blaSPM, which is one of the metallo-beta lactamases distributed in Gram-Negative pathogens.

Reply: The abstract was modified as requested.

  1. In the introduction, L55, I recommend changing “between bacteria” to “among pathogens).

Reply: The text was modified as suggested.

  1. In the introduction, I recommend including some information about carbapenemases other than MBL and the criteria of each one as well as more elaborative information about blaSPM and whether it is plasmid-meditated or not with citation of relevant

Reply: The text was modified as suggested.

  1. L102-103, the authors should provide the URl of each online website together with the accession date for each

Reply: The website address for each database was included as requested.

  1. L56, 57, the author should add. (It is worth mentioning that bacteria co-expressing SBLs and MBLs are usually able to hydrolyze the clinically relevant monobactam, aztreonam with citation of the relevant literature such as. https://doi.org/10.2144/fsoa-2019-0098; https://doi.org/10.1016/j.tim.2011.09.005

Reply: The text was modified as suggested.

  1. The abbreviations should be mentioned in full at first mention, the whole manuscript should be accordingly

Reply: The abbreviations were revised and the text modified as suggested.

  1. P6, what is the rationale for the yellow highlighting of Santos et

Reply: The reference was flagged, as we were awaiting the publication of the article: Endemic High-Risk Clone ST277 Is Related to the Spread of SPM-1-Producing Pseudomonas aeruginosa during the COVID-19 Pandemic Period in Northern Brazil (Santos et al. 2023), for inclusion of the appropriate reference. (https://www.mdpi.com/2076-2607/11/8/2069).

  1. L168, aeruginosa should be italicized. The whole manuscript should be revised accordingly.

Reply: The text modified as suggested.

  1. Figures 2-7, I recommend including better-resolution

Reply: The figures were modified was requested.

  1. I recommend including a section in the results for comparing the prevalence of MBL in Brazil to that of other geographical areas worldwide particularly those of similar standards of living and in order to have a picture of the obtained percentage in relation to other

Reply: Comparison data as added as requested.

Therefore, and for the above-mentioned remarks, I advised a minor revision of the respective manuscript in its current state taking into consideration the above comments and recommendations before being considered for publication.

We appreciate the valuable considerations of the reviewer, which have been included in the manuscript in its entirety.

Reviewer 2 Report

The authors have conducted an exhaustive review of the prevalence of metallobetalactamase-producing Pseudomonas aeruginosa isolates in Brazil. The topic is particularly relevant because of the difficulties of treatment that these microorganisms present in addition to the probability of horizontal transmission leading to outbreaks. It is very appropriate to stratify the prevalence by the different territories analysed in that country. Although we may think that this is a local problem in the country where the study was carried out, it is very important to publish these data for general knowledge. Globalisation and migratory movements mean that multi-resistant micro-organisms could affect other countries in the future, and doctors should be aware of this information in order to take the necessary control measures. This is why this publication is of general interest and meets the criteria for publication. Both the substance and the form are correct.

Author Response

We greatly appreciate the positive comments on the manuscript.